# Patient coaching: What do patients want? A mixed methods study in waiting rooms of outpatient clinics

Irène Alders[1]*, Carolien Smits[2], Paul Brand[3,4], Sandra van Dulmen[1,5,6]

1 Department of Primary and Community Care, Radboud Institute for Health Sciences, Radboud University Medical Center, Nijmegen, Netherlands, 2 Program Older Adults and Health, Pharos, Dutch Centre of Expertise on Health Disparities, Utrecht, Netherlands, 3 Department of Innovation and Research, Isala Hospital, Zwolle, The Netherlands, 4 Postgraduate School of Medicine, University Medical Centre Groningen, Groningen, Netherlands, 5 Nivel (Netherlands institute for health services research), Utrecht, Netherlands, 6 Faculty of Caring Science, University of Borås, Borås, Sweden

◉ These authors contributed equally to this work.
* iren.alders@radboudumc.nl

**Data Availability Statement:** All relevant data are within the paper and its Supporting Information files.

**Funding:** The authors received no specific funding for this work.

## Abstract

### Introduction

Effective communication in specialist consultations is difficult for some patients. These patients could benefit from support from a coach who accompanies them to and during medical specialist consultations to improve communication in the consultation room. This study aims to investigate patients' perspective on interest in support from a patient coach, what kind of support they would like to receive and what characterizes an ideal patient coach.

### Methods

We applied a mixed method design to obtain a realistic understanding of patients' perspectives on a patient coach. Patients in the waiting rooms of outpatient clinics were asked to fill out a short questionnaire which included questions about demographic characteristics, perceived efficacy in patient-provider interaction and patients' interest in support from a patient coach. Subsequently, patients interested in a patient coach were asked to participate in a semi-structured interview. The quantitative data were examined using univariate analysis and the qualitative interview data were analysed using content analysis.

### Results

The survey was completed by 154 patients and eight of them were interviewed. Perceived efficacy in patient-physician interactions was the only variable that showed a significant difference between patients with and without an interest in support from a patient coach. The interviews revealed that a bad communication experience was the main reason for having an interest in support from a patient coach. Before the consultation, a patient coach should take the time to get to know the patient, build trust, and help the patient create an agenda, so take the patient seriously and recognize the patient as a whole person. During the consultation, a patient coach should support the patient by intervening and mediating when

**Competing interests:** The authors have declared that no competing interests exist.

necessary to elicit the patient's agenda. After the consultation, a patient coach should be able to explain and discuss medical information and treatment consequences. An ideal patient coach should have medical knowledge, a strong personality and good communication skills.

## Conclusion

Especially patients who had a bad communication experience in a specialist consultation would like support from a patient coach. The kind of support they valued most was intervening and mediating during the consultation. To build the necessary trust, patient coaches should take time to get to know the patient and take the patient seriously. Medical knowledge, good communication skills and a strong personality were considered prerequisites for patient coaches to be capable to intervene in specialist consultations.

## Introduction

Effective communication in medical consultations is positively associated with treatment adherence, decreased medication errors and stronger patient engagement, for instance in shared decision-making [1]. In shared decision making the contribution of a patient is essential. However, not all patients are able to communicate effectively in consultations with medical specialists. This is caused by the patient's emotional state, like feeling tense or overwhelmed, the perceived time pressure, uncertainty about their own understanding, not wanting to be bothersome, remembering questions only after the consultation and also the attitude of the professional [2–4]. Furthermore, patients are hindered by the power imbalance [5], or their inability to change the agenda in the consultation [6]. Although medical specialists are increasingly trained in communication skills, transfer to real consultations is still limited [7]. Furthermore, the consultation time remains limited, and training medical specialists does not solve the experienced power imbalance. To support patients in communicating effectively during these consultations, several guiding and coaching interventions for patients have been developed and investigated [8, 9]. It appears that *personal*, face-to-face support may be best suited. The human connection is invaluable in the context of person-centered care and helps to make patients feel respected and equal [10]. When a patient coach spends time with a patient in preparation of the consultation(s), he gets to know the patient in his own context. During the accompanied consultations, the patient's specific communication barriers are enlightened and can be addressed by the patient coach. Personal support can easily and instantly be adjusted to better meet an individual patient's circumstances and needs [2].

One of these patient-directed personal interventions is patient coaching. We defined the concept of patient coaching as personal support for patients, aiming at improving communication in consultation with a medical specialist. The patient is supported in the preparation of the consultation, accompanied during the consultation and in the evaluation of the consultation with a medical specialist afterwards. Such an intervention can sustainably improve communication effectiveness during, immediately after and even weeks after the consultation [8]. However, whether patient coaching is effective only becomes apparent during a consultation, so it is important that the coach accompanies the patient in the consultation room, observes the situation and intervenes with support "in action" if necessary [11]. The only intervention so far that is comparable to our concept of patient coaching is Consultation Planning

Recording and Summarizing (CPRS), an intervention in which a patient coach ("navigator") accompanies a patient and remains in the room during the consultation. Research on CPRS, however, mainly focusses on the effects of support on decisional satisfaction [11–16] rather than patients' needs and preferences. There still seems to be a gap between the support patients expect and need from a patient coach and the actual support they receive.

Patient coach interventions have been studied in patients with a variety of clinical and demographic characteristics [8]. A retrospective study of Dutch outpatients with a chronic disease showed that one in six patients would like support from a patient coach when consulting a medical specialist [3]. Further analysis showed that patients with and without interest in a patient coach differed on three specific communication barriers. Patients interested in a coach (a) were too nervous to ask questions, (b) doubted whether the specialist in question was the right person to answer their questions, or (c) were uncertain about their own understanding, leaving them with unsolved questions and worries [4]. More in-depth insight in which patients actually want support is necessary to attune a patient coach intervention.

We previously investigated healthcare professionals' views on patient coaching. Healthcare professionals from different backgrounds and experience think that patients who are vulnerable, either generally or situationally, might benefit from communicative support from a patient coach when consulting a medical specialist [17]. Generally vulnerable patients were characterized as older, as patients with impaired cognition, an insufficient support network or a lower level of health literacy. Situationally vulnerable patients were described as patients who are anxious due to the situation, which is independent of age, educational level or level of health literacy [17]. Although most patients do not consider themselves vulnerable [18], we expect that patients will be able to imagine the kind of personal support that benefits them in consultations with specialists just before a consultation begins.

The type of support patients prefer determines the patient coach profile. So far, research on patient coaches has shown that they have various backgrounds, ranging from lay educators to trained professionals, but a relationship between the coach's profile and goals or outcomes of coaching has not been investigated [8]. In a US study on breast cancer patients, two types of CPRS patient coaches were compared: schedulers and premedical interns [14]. This study found that physicians endorsed CPRS as it supported patients in preparing questions and ensured that answers were given during the consultations. The participating physicians suggested to deploy clinical research assistants as patient coaches without providing a reason for their suggestion [14].

Patients are often accompanied by family or friends during medical consultations to support them, however, their emotional involvement and relationship may conflict with the patient's needs [19]. It still remains unclear whether patients prefer a family member or a professional coach to support them during consultations, whether a coach should have medical knowledge and which particular skills are valued.

To shed more light on individual patients' needs, we investigated the characteristics of patients that would like support from a patient coach when consulting a medical specialist and their reasons for the desired support. Subsequently, we combined patients' preferences for support with their views on a coach profile to explore what characterizes the ideal coach. This, in turn, may provide valuable information for the design of a training program.

Our research questions were: which patients are interested in support from a patient coach, how should a patient coach support a patient, and what characterizes the ideal patient coach?

## Methods

This mixed methods study comprised a survey amongst patients in an outpatient clinic waiting room and subsequent in-depth interviews with a sample of the survey respondents who had

indicated an interest in a patient coach [20]. A mixed methods design was chosen to obtain a more profound understanding of patients' perspectives on support from a patient coach.

## Context, participants, and ethical considerations

The study was conducted in 2018 at Isala, a large general teaching hospital in the Netherlands. We invited 203 patients in the waiting room of outpatient clinics for chronic diseases (cardiology, pulmonology, rheumatology, oncology, internal medicine, and geriatrics) to participate in our study. In these clinics, we were likely to encounter vulnerable patients. Prior to a consultation with a medical specialist, two Bachelor's of Nursing students informed the patients about the objectives and procedures of the study. They explained the concept of patient coaching and asked the patients to participate in the survey. Their participation was voluntary, and they could withdraw at any time. All participating patients in the survey provided informed consent. For the interviews, seven patients provided written consent. The interview with one patient was cancelled in agreement with the patient's partner, because of the patient's cognitive condition. A day later this interview was continued by his partner by telephone in which additional oral consent was obtained and audio recorded. The eighth participant preferred to participate in the interview by telephone. This informed consent was obtained orally and audio recorded. The study was approved by the Medical Ethics Committee of Isala hospital, a licensed subsidiary of the Dutch national Medical Ethics Review Board (number: 180339).

## Study design

To answer our first research question, patients were asked to complete a short, tailor-made questionnaire, which was given to them by the nursing students prior to a consultation with a medical specialist (S1 Appendix). In addition to questions on demographic, education and disease-related information, patients were asked to estimate their efficacy in patient-physician interactions using the validated PEPPI-5 questionnaire (Perceived Efficacy in Patient-Provider Interaction, short questionnaire) [21]. The PEPPI-5 includes five items that have to be rated on a 5-point Likert Scale, ranging from very confident to not confident at all. Furthermore, patients were asked to indicate whether they were interested in a patient coach and explain their choice (open text field). Interested patients were asked whether they were willing to take part in an interview.

In addition to the information gathered from the survey, we took a convenience sample of patients who had affirmatively answered the questions on suffering from a chronic disease, interest in a patient coach and willingness to participate in an interview. Our aim was to supplement the quantitative results on the first research question and explore patients' experiences and perceptions to seek an answer to the second and third research question. Two nursing students were trained in interview techniques. Subsequently they interviewed the participants guided by a topic list (S2 Appendix), which was based on the topic list of our previous study among healthcare professionals [17]. During the interview, patients were asked to explain why they were interested in support in consultations with medical specialists, what kind of support they would prefer and who they think would be best suited to provide the support they need.

The interviews were conducted at the patients' homes at a time convenient to them, within two weeks after the consultation. The concept of patient coaching was, again, explained during the introduction of the interview and illustrated with a short animation (https://youtu.be/iF4kkHG2l2M) (Reprinted from Youtube under a CC BY license, with permission from Irène Alders, original copyright 2018).

### Data collection and analysis

Data were collected in April and May 2018. For analysis and interpretation purposes, patients' diseases were clustered in disease groups and the scores of patients' perceived efficacy in patient-provider interaction were dichotomized into low confidence (scores 1–3) and high confidence (scores >3). Quantitative data were analysed using SPSS 25. Differences in characteristics between patients with and without an interest in a patient coach and patients' perceived efficacy in patient-physician interactions were examined using univariate analyses (Pearson's chi-squared and Fisher's exact test (1-sided)). For the interviews, we followed the "Standards for Reporting Qualitative Research" (SRQR) recommendations [22]. The interviews were audio-recorded, transcribed verbatim by the nursing students and subsequently all checked for accurate transcription by the first author. The transcripts were analysed through content analysis [23]. First, the first and second author read the transcripts to familiarize themselves with the data. Then the nursing students (by hand) and the first author (in Atlas ti) independently open-coded the transcripts. The first author made excerpts of the transcripts. Subsequently, the first and second author discussed the codes, categories and themes resulting from the coding process and the excerpts, until agreement was reached. The final agreement on codes, categories and themes was reached after repeated discussions within the entire research team.

### Research team and reflexivity

It is inevitable that the researchers' prior experiences, assumptions and beliefs influence the research process, therefore, reflexivity is essential [24]. The diversity of our interdisciplinary research team added to the rigour and quality of the research, increased creativity, and intellectual rigour, and helped reduce researcher bias. The first author (IA, female) is very familiar with patient coaching because she herself had worked as a patient coach to test the concept. This may have affected the way she interpreted the transcript data. The nursing students (both female) were doing a research project for the first time and were recently trained in interviewing techniques. They had no experience in patient coaching. The first author mentored their research. The second author (CS, female) is a geronto-psychologist and senior researcher with considerable expertise and experience in qualitative research. This guided the decision to define and discuss the themes and categories with her. The other researchers, a psychologist (female) and a medical specialist (male), added a broader perspective to the research/ discussions.

## Results

In total, 203 patients were asked to participate in the survey and 154 of them completed the questionnaire (76%). Twenty-one patients (13.6%) were interested in support from a patient coach and 11 of them agreed to participate in an interview. Three patients were not able to make an appointment within two weeks after the survey. No further data were collected on reasons for non-participation.

Characteristics of patients with and without an interest in a patient coach are presented in Table 1. Patients' age (above and below 65 years) and sex were equally represented in the survey sample. Most patients had a lower or medium level of education and were accompanied to the consultation, mostly by their partner.

Eight patients were interviewed, two of them together with their partners (P2 and P2Partner, P4 and P4Partner). One accompanying friend (with a paramedical background) was interviewed on behalf of a patient who was suffering from mild cognitive impairment (P5). Two

**Table 1. Characteristics of the patients who completed the questionnaire.**

| | | total (n = 154) | interested in a coach (n = 21) | not interested in a coach (n = 133) | p-value |
|---|---|---|---|---|---|
| **Age** | Mean (SD) | 62.3 (16.4) | 59.6 (18.5) | 62.7 (16.1) | |
| | | | % | % | |
| | < 65 Years | 78 | 57.1 | 49.6 | .343^^ |
| | > = 65 years | 76 | 42.9 | 50.4 | |
| **Sex** | | | | | |
| | Male | 72 | 42.9 | 47.7 | .442^^ |
| | Female | 82 | 57.1 | 52.6 | |
| **Education** | | | | | |
| | Low | 51 | 52.4 | 30.1 | .130^ |
| | Medium | 71 | 33.3 | 48.1 | |
| | High | 32 | 14.3 | 21.8 | |
| **Type of disease**\* | | | | | |
| | Cardiovascular | 16 | 18.8 | 16.7 | .755^ |
| | Respiratory | 8 | 12.5 | 7.7 | |
| | Musculoskeletal | 26 | 25.0 | 28.2 | |
| | Cancer | 21 | 18.8 | 23.1 | |
| | Endocrine | 15 | 18.8 | 15.4 | |
| | Neurological | 2 | 6.3 | 1.3 | |
| | Other | 6 | 0.0 | 7.7 | |
| **Comorbidity** | | | | | |
| | Yes | 25 | 23.8 | 15.0 | .235^^ |
| | No | 129 | 76.2 | 85.0 | |
| **Outpatient clinic** | | | | | |
| | Cardiology | 39 | 23.8 | 25.6 | .100^ |
| | Pulmonary medicine | 25 | 14.3 | 16.5 | |
| | Rheumatology | 21 | 4.8 | 15.0 | |
| | Oncology | 25 | 19.0 | 15.8 | |
| | Hematology | 12 | 9.5 | 7.5 | |
| | Internal medicine | 16 | 0.0 | 12.0 | |
| | Internal medicine, nephrology | 4 | 4.8 | 2.3 | |
| | Internal medicine, diabetes | 6 | 9.5 | 3.0 | |
| | Geriatric medicine | 6 | 14.3 | 2.3 | |
| **Accompanied** | | | | | |
| | No | 56 | 38.1 | 36.1 | .520^^ |
| | Yes | 98 | 61.9 | 63.9 | |
| **Accompanied by whom** | | | | | |
| | Alone | 57 | 38.1 | 36.8 | .277^ |
| | By a partner | 65 | 42.9 | 42.1 | |
| | By a child | 17 | 9.5 | 11.3 | |
| | By a friend | 7 | 4.8 | 4.5 | |
| | By a professional | 1 | 4.8 | 0.0 | |
| | By a parent | 5 | 0.0 | 3.8 | |
| | Other | 2 | 0.0 | 3.8 | |
| **Perceived efficacy (PEPPI) score, Mean (SD)** | | | 4.0 (1.1) | 4.5 (.6) | |
| | | | % | % | |
| | Low self efficacy (1–3) | 10 | 33.3 | 2.3 | .000^^ |

(*Continued*)

**Table 1.** (Continued)

| | | total (n = 154) | interested in a coach (n = 21) | not interested in a coach (n = 133) | p-value |
|---|---|---|---|---|---|
| | High self efficacy (>3) | 144 | 66.7 | 97.7 | |

Sample size was 154, except for * type of disease: 60 missing cases

SD = standard deviation; PEPPI = Perceived efficacy patient provider interaction

^ Pearson chi square

^^ Fisher's exact test (1-sided)

interviews were conducted by phone. We stopped data collection when our data sufficiently satisfied our exploratory research questions. Interviews lasted on average 50 minutes.

Characteristics of patients interested in a coach

## The survey

Patients who were interested in a patient coach more often reported lower efficacy in interaction with their medical specialist than patients with no interest. We did not identify any other factors that could be associated with an interest in patient coaching. The results did not differ when age was used as a continuous variable (p = 0.156, Pearson chi square, 2 sided, range 19–87 years) (Table 1).

Patients' main reasons for interest in support from a patient coach were to help them remember questions during the consultation, to ask better questions, to feel more self-confident, to improve interaction with the physician and to accompany them when family is unavailable. The most common reasons for patients who did not express an interest in support from a patient coach were experiencing no problems in the consultation, feeling sufficiently effective in interaction with their medical specialist and having family members available to accompany them.

## The interviews

The interviewees were largely representative of the survey population with respect to age, sex, and educational level. All but one patient reported that they were usually accompanied to consultations by family members or friends, except for follow-up visits. Three patients suffered from cognitive impairment due to dementia or an accident. At the time of the interview, one patient was diagnosed with cancer and most patients suffered from multiple diseases, had an illness duration of more than six years, and had consulted more than one medical specialist. Table 2 shows the themes and categories we identified based on the interview data. We provide an example quotation for each category and present the remaining quotations when eloborating on the identified categories.

*Communication experiences.* The interviewees reported that they considered themselves effective communicators in consultations with their medical specialist. The main reason for interest in a coach was that they had a bad communication experience and felt dissatisfied after consulting a specific medical specialist. The interviewed patients or their relatives felt that the medical specialist did not take their views and experiences seriously or experienced a lack of empathy. The patients were unable to effectively address the effects of the specialist's communication style during the consultation, leaving them with feelings of anger and frustration.

> "*. . . . he [the medical specialist] had one [such a device] himself and it worked well for him. He couldn't understand that it wasn't good for me. . .. no, the doctor didn't understand that it wasn't good for me.*" P2

**Table 2. Themes, categories, and example quotation.**

| Theme | Category | Example quotation |
|---|---|---|
| **patient characteristics** | communication experience | *"I only think doctor [name]. . . I'm devastated about that. . . . That's our geriatrician. I think it's a. . . Let me say. . . less pleasant man. Let me put it this way: I would be very happy if some kind of coach accompanied us. I do not intend to go there again. I don't want to go there anymore." P4Partner* |
| | disease stage: in a trajectory | *. . . for the specialist for my mouth I never need a coach. . . I have been treated for 30 years in special dentistry." P3* |
| | | *"It is really exactly during that time that you are sick and you hear everything, because then you are left with unanswered questions because they simply don't have time. . .Now I'm in a traject of post-checks and controls. I don't think that it is important to be accompanied by someone like a coach. . ." P6* |
| | lack of suppport system, assertiveness or cognitive capacity | *"For myself I would also say that if I had a medical problem and my daughter couldn't [accompany me], . . .Or my son-in-law could not, I think that a coach, as I read, would be very positive." P3* |
| | | *"Well, I can well imagine that when you are alone and you aren't hundred percent [mentally healthy] anymore or you dare not speak your mind, then a coach would be ideal." P5* |
| **preferred kind of support** | before and between consultations: preparation and buliding trust | *"If there is no trust, I don't think it makes much sense to act as a coach." P5* |
| | | *"And very often patients don't know what causes it [the medical problem] and then it is the coach's skill to figure out where the medical problem is coming from. By asking very specific questions and, what I now notice in people suffering from dementia, that this can be very difficult. . . Well, to really get the right information from someone." P5* |
| | during consultation: improve patient-physician interaction | *"So the doctor asks me questions about this and that, that's actually normal. But maybe I forget something, that the coach will remember." P3* |
| | after consultation: explain and discuss medical information | *"He actually needs to know a little bit more about everything around [the illness]. Of the things around it. The consequences." P2* |
| **patient coach profile** | medical knowledge | *"It has to be someone who listens to you and who also provides answers.". . . . "Someone who knows what it's all about and who can see for himself, oh yes, you have to ask or talk about this as well. Who is a bit more in control." P6* |
| | communication skills | *". . .He or she also functions a bit as a buffer. So to prevent that when you return home you are very aroused and angry. . . If someone guides that process and who can also defend himself against a doctor more easily and say: 'hey, something is going wrong here.'" P4Partner* |
| | strong personality | *"That the coach has sufficient influence to say: 'Mister or doctor now you are moving a bit in the wrong direction'. . . . Who has insight, but also authority.. . ." P4Partner* |
| | professional or relative | *"Well, then I'm also worried about what will happen. I'm just worried about it. Then I think 'He's going to work too hard again.'" P2Partner* |

> *"When I was told that I had cancer, I was alone [in the room with the specialist]. And it went in such a way. . . that I thought, well . . . That man put on his coat, and he said it and then walked away. And then you're left with a lot of questions. At that moment, your world collapses. After that experience, my wife has always accompanied me. Anyway, you always get stuck with things. . . with that first man. . .. That was a jerk first class, let's put it that way." P6*

The interviewees also mentioned several communication problems such as not being listened to, or physicians distracted by their computer screen. Several interviewees expressed lack of being recognized as a whole person, with co-morbidity, history, and experience in healthcare as a reason for their interest in a patient coach.

*Disease stage.* Patient's stage of disease and treatment did not seem to affect their interest in support from a coach, apart from a slight preference for support shortly after the diagnosis of cancer or diabetes. This wish for support mainly concerned coping with the diagnosis, disease, treatment, and consequences.

> *"Maybe in the beginning when I got ill and when I was hospitalized. Maybe you see him [the patient coach] once a day, or once a week, or twice, or when you're worried about something*

*or when you have a hard time. . .. being able to call someone like that . . .. But right now? No, now I don't need it." P6*

*Lack of support system, assertiveness, or cognitive capacity.* Most interviewees imagined that support from a patient coach would be very welcome for patients other than themselves. They suggested a patient coach for patients who lack a support system, who are less pro-active/assertive, and cognitively impaired patients.

## Preferred kind of support

**Support before, during and after consultations.** *Before and between consultations*: interviewees felt that a patient coach should take time to get to know them and build trust. *During consultations*: most interviewees emphasized a need for support to influence the communication with the medical specialist and support in question asking. *After consultations*: interviewees would like to discuss medical information and consequences of treatment decisions with the coach. *Follow-up consultations*: the interviewees reported not to need patient coaches to accompany them to follow-up consultations, but remarked that should depend on the patient's preferences. Patients and coaches should make good arrangements about how and when to contact each other.

**Build trust and a bond.** Participants stressed the importance of building a trustful relationship with a patient coach. They felt trust can be built up between patient and coach through multiple contacts, also between consultations. A patient coach should be approachable and available upon request by phone, email or in person, e.g., after a consultation with a medical specialist when explanation of medical information is needed or a patient has further questions. Interviewees would prefer to be visited at home, allowing the patient coach to observe the patient's home situation.

*"You see, you are building a bond with someone, so yes, that would be once or twice a month or so. Or make a phone call. . . And that creates trust, and that's what it's all about. That bit of confidence." P6*

Most participants expressed a preference for a single coach, so they did not need to repeatedly explain their situation. One participant would like a clear offer in coaches, a list of available coaches to choose from.

**Prepare the consultation, agenda setting.** A patient coach can support a patient before the consultation by making an inventory of the medical issue(s), medication/treatment issues, questions, and concerns. This inventory should also include the patient's care history, experiences, context (hobbies, responsibilities, social network), to get a clear picture of the patient's overall situation. Subsequently, based on the inventory the patient coach and the patient can prepare the patient's agenda for the consultation in advance.

*"You have to talk to that man [the patient coach], or woman, it doesn't matter what gender the coach has. And what you want to talk about. And when you talk about your heart, it's also about medication, but you can also have other questions. And that you discuss with your coach: I want to discuss this [issue] and that [one during the consultation with the medical specialist]." P3*

**Improving patient-physician interactions.** During the consultation, a patient coach should keep oversight and ensure that the patient gets the opportunity to provide all relevant contextual information, even when a medical specialist does not sufficiently facilitate this.

Patients emphasized that they would prefer to speak for themselves, with the patient coach only stepping in when they forget to provide essential information or to ask prepared or other relevant questions.

> *"You see, you are facing a lot of things and some people in this area, just know a lot. They [patient coaches] also know that someone will shut up at some point and doesn't know what to ask anymore. They [patient coaches themselves] will start a conversation, so that at some point you will be able to ask and say things again. . . .. who also know that you will shut up [when talking about] some things and then get you over the hump by asking questions or restarting a conversation. . . who understand what you want to ask or talk about. Who control [the dialogue] a bit more."* P6

Interviewees also felt that a coach can mediate if a patient (or a family member) becomes emotionally upset or has the feeling of not being taken seriously.

> *"Because when the family or the patient is not listened to, the coach can intervene . . .. A coach can exert more influence on this [process]. Say, [influence] to both the doctor and the patient."* P5

A patient coach could support a patient in checking, probing, and recording medical information to ensure understanding and recalling.

> *"That [the patient coach] is indeed observing the patient, does the patient understand it in the end too, say, when he's at home?. . . . Because he listens in [during the consultation], checks whether everything has been discussed or whether there are any questions left."* P8

**Explain and discuss medical information.** After the consultation, a patient coach should explain and discuss the medical information and its consequences on daily life and support a patient emotionally.

> *"It's actually exactly in the period that you are also sick and hear all sorts of things, because then you are left with questions that cannot be answered, because they simply don't have the time for it."* P6

When specifically asked about the need for emotional support during the consultation, one interviewee stated that his wife covered this need.

## Patient coach profile

**Medical knowledge and communication skills.** Medical knowledge was generally considered a prerequisite for being able to empathize with a patient, keep oversight and influence the communication process during the consultation. A patient coach should be able to explain and discuss medical information and consequences of treatment choices afterwards.

> *"That he has knowledge about the subject, of course. That he has knowledge about dementia, Alzheimer's, MCI. All those milder forms too. . . .. That apnea causes so much oxygen deficiency, that you can start to feel woozy and that you can't drive a car anymore. That he [the patient coach] knows what he's talking about."* P4Partner

According to the interviewees, specific communication skills to build trust, maintain trustful relationships and get to know the patient on a personal level are good listening and

observation skills. To have a positive effect on the communication process during the consultation, a patient coach should also be able to intervene and mediate when patient's agenda is insufficiently elicited.

> *"But if the specialist sends the patient away [home] without really looking at the current medical problems, then I think a coach should intervene." P5*

One participant raised the issue of specialized coaches for each type of disease. These specialized coaches would then have to exchange information and one of the coaches would act as the patient's contact.

**Personality and other skills.**    Participants agreed that the main characteristics of a patient coach should be taking the patient seriously, being kind and being trustworthy. Furthermore, a strong personality may be needed to manage the conversation during consultations.

> *"I think that a coach has to be a strong person, who listens well to people [others], who is also aware of his own limits, but who also knows what the limits of the patient are. And who can effectively deal with someone [complex people and situations]. That's also* important." P8

**Professional or relative.**    Although most participants were usually accompanied by relatives, they were still interested in support from a patient coach. They expected a patient coach to be better able to persuade a medical specialist to be more receptive to their reasoning when the patient coach has medical knowledge and is not emotionally involved.

A patient coach would be a good alternative when family members are not available. One interviewee felt burdened to ask for support from a patient coach because her family was able to support her, but she would like a patient coach as well.

> *"Look, I want to spare my daughter [the stress of accompanying me to the hospital], because she's just very busy. . . But I would offend her if I had a coach, because then she would say: 'Mommy, we're here for you.'" P1*

## Discussion

In this study we investigated the views of patients in the waiting room of outpatient clinics on patient coaching to support effective communication in consultations with a medical specialist. First, we used a survey to develop a broad scope on patients' views on patient coaching. Subsequently, within two weeks, we conducted in-depth interviews to allow a detailed exploration of individual patients' interest in support from a patient coach. The survey showed that one in seven patients was interested in support from a patient coach, mostly when family members were not available to accompany them. Perceived efficacy in patient-physician interactions was the only variable that showed a significant difference between patients with and without an interest in support from a patient coach. The interviews showed that patients' main reason for having interest in a patient coach was that they had a bad communication experience. The interviewed patients would like support in preparing their own agenda for the consultation. Patient coaches were perceived to be most important for effective communication during the consultation, which would not be possible without proper preparation. In the preparatory phase, which includes preparing for the consultation and maintaining contact between consultations, trust could be built between the patient and coach. During the consultation, the interviewees mostly preferred to be supported in managing the conversation with

the medical specialist, so they would feel taken seriously and heard. After the consultation, the interviewees would like to discuss the medical information with the patient coach, who should be able to help them process the information and explain the consequences of treatment choices. To be able to provide this kind of support an ideal patient coach should take time to get to know the patient, become familiar with the patient's (medical) history and circumstances, be a kind, trustworthy professional with sufficient medical knowledge and good communication skills, and have a strong personality.

A bad communication experience with a specific medical specialist was an important reason for patients to have an interest in a patient coach. As the medical consultation plays a central role, poor communication can negatively influence patients' perceptions of the quality and effectiveness of healthcare [25]. The interviewees felt not taken seriously by the specialist and were not able to change that situation on their own. The feeling of not being taken seriously is related to the relationship with the medical specialist and not being able to exert influence is related to communication skills. Although healthcare professionals have increasingly been trained in effective communication skills, patients still report barriers to communicating effectively with doctors [2, 3, 26–28]. From a healthcare professionals' perspective, effective medical communication has six functions: (1) fostering the relationship, (2) gathering information, (3) information provision, (4) decision making, (5) enabling disease and treatment-related behaviour, and (6) responding to emotions [29]. According to the experiences of the interviewees, at least the first two functions were insufficiently addressed by the medical specialists and in some medical consultations, neither was the sixth function.

Fostering the relationship, the first function of effective communication, is an essential basis for quality of care. The absence of such a relationship or a poor relationship between the patient and medical specialist may lead to withdrawal of care, non-adherence, misunderstandings, dissatisfaction, formal complaints, or medical errors [30, 31]. Essential elements for fostering the physician-patient relationship are building respect, trust, and rapport, which all contribute to the patient's feeling of being known [29]. The interviewees would like support from a patient coach comprising these essential elements when not provided by the medical specialist, which could be seen as a substitute for the medical specialist's time and attention. This support should be given in person, because a process of human connectedness appears to be more valued by patients, than the provision of written or online information only [32].

Although patients do not have the main responsibility for effective communication in the consultation, they do have an active role in the physician's information gathering phase of the consultation, the second function of effective medical communication: patients need to provide relevant information. Patients who more actively participate in the consultation tend to get more targeted information [33, 34]. To be able to provide relevant information, patients need to prepare the consultation. They have to create an agenda by prioritizing their goals, questions, and concerns to establish focus on their needs [35]. Subsequently, during the consultation, patients need to be able to elicit their agenda [36]. Despite the need to be taken seriously and heard in a consultation, only one in ten patients saw an active role for themselves in preparing a consultation [37]. Next to personal support, several non-personal preparatory interventions have been offered to patients to support them, for example online or printed decision aids, question prompt lists and communicative support like PatientVOICE or PatientWisdom [38, 39]. However, these interventions mainly prepare the patient for the decisional phase of the consultation, which might lead to different outcomes than assessing patients' concerns [40]. Although these interventions are valued by patients, their use is still limited [8, 38, 41, 42].

In the personal CPRS intervention, in which a patient coach accompanies patients to a consultation, the patient coach only records and summarizes the consultation and does not

interrupt or redirect the conversation [12, 14]. In approximately four out of five patient-specialist encounters patients did not get the opportunity to discuss their agenda [36]. In most consultations, patients were interrupted after a median of 11 seconds [36]. On top of that, patients may express concerns and emotional issues in a subtle way, which can be easily overlooked by the medical specialist [43]. It requires high levels of interactive self-efficacy, such as keeping oversight, intervening abilities or clearly expressing wants or needs, to redirect the conversation towards the patients' agenda in the limited time available, which cannot be provided by support in preparing the consultation alone [5, 44].

In intervention studies, vulnerability is mostly associated with perceived communication barriers, older age, lower levels of health literacy, absence of a social network and severity of illness [3, 4, 45]. Our previous research showed that healthcare professionals could imagine that generally or situationally vulnerable patients might benefit from support from a patient coach [17]. In the present study, we presumed that patients would not consider themselves vulnerable [18, 46], but might want support from a patient coach when they were sitting in the waiting room, facing a consultation. The patient's vulnerability was apparent in the interviews, as patients described their inability to effectively communicate with their medical specialist. This kind of vulnerability depends on several factors, like a patient's personal characteristics and situational factors, leading to a mismatch between the patient's needs and the healthcare provided [47]. The different communication stages patients encounter during their patient journey can be identified as being: (1) overwhelmed, passive, (2) pro-active, self-motivated, and (3) proficient, empowered. When the time passes after the diagnosis and patients become more experienced and less overwhelmed, most of them have learned to be more pro-active and are more empowered in consultations with their medical specialist [2]. However, every new shocking test-result may cause a throw-back into an overwhelmed stage, making the patient vulnerable again and in need of communicative support.

Our findings reflect the results of our previous study aimed at characterizing patients with an interest in a patient coach using a survey among members of a patient panel. Contrary to our patient sample, the surveyed patients in the previous study were not awaiting a specialist consultation. They had to imagine a situation where support from a coach could be helpful or beneficial, which might explain differences in responses. Communication barriers that distinguish patient with and without an interest in a patient coach are feeling tense, feeling uncertain about one's own understanding and believing that a certain topic is not part of the specialists' responsibilities [4]. Since our interviewees were interested in a patient coach after a bad communication experience and patients do not express their need for patient coach themselves [18], involved healthcare professionals need to be alert to patients' signals that could be indicative of experiencing barriers to effective communication. Possibly, patient coaches can help overcome barriers and prevent bad communication experiences.

Some of the interviewees missed support in coping with their disease. However, our concept of patient coaching only focusses on patient-specialist communication in consultations, since support from a patient coach may improve effective communication "in-action". It needs to be clear for both patients, medical specialists and other involved healthcare professionals what can be expected from a patient coach. Patients may be reluctant to discuss their need for additional support with their medical specialist, because of their hindering belief that it is not the responsibility of this physician to discuss a specific topic [4]. A patient coach could support patients by managing their expectations on what they can ask in a consultation and enabling them to express it, and when expressed, giving the medical specialist the opportunity to address it.

### Strengths and limitations

A strength of this study is the mixed methods design. We combined a survey with general questions with in-depth interviews to explore patients' views and needs for patient coaching. Furthermore, patients were asked to participate when they were in the waiting room before having to confront an arousing situation, which could have helped them imagine what kind of support they would like. On the other hand, facing a consultation could have triggered a coping mechanism causing overestimation of their own interaction efficacy. Limitations of our study are that we recruited patients from a single hospital and interviewed a small number of patients, which may limit generalizability.

## Conclusions

Especially patients who had experienced a consultation in which they felt not being taken seriously or heard by their medical specialist were interested in support from a patient coach in future consultations. A patient coach should help the patient prepare the agenda *before the consultation*, accompany the patient to a consultation and ensure that the main agenda items are discussed *during the consultation* and discuss the consequences of treatment choices *after the consultation*. Medical knowledge, good communication skills and a strong personality are prerequisites for a patient coach to be able to intervene in a consultation if necessary and explain the consequences of treatment choices.

### Future research

Future research should explore whether our findings can be generalized to other patient populations and other settings. Further research should also be done to understand which training of befitting patient coaches realizes achievement of the desired outcomes of patient coaching.

### Practical implications

Healthcare professionals should be alert to ineffective communication when patients mention bad communication experiences, barriers to talk to a medical specialist or show signs of general or situational vulnerability. These patients may need communicative support and as long as the role of a patient coach is not yet officially established, healthcare professionals should strongly recommend the patients to bring support from their social network. To provide professional patient coaches to vulnerable patients, candidates who fit the profile need to be sufficiently trained.

## Supporting information

**S1 Appendix. Appendix I survey questionnaire.**
(PDF)

**S2 Appendix. Appendix II topic list interviews.**
(PDF)

## Acknowledgments

We like to thank Nancy Stuivenvolt and Marsha Prinsen for their contribution to data collection and analysis in the context of their research minor in their study Nursing. Furthermore, we like to thank the students of the Deltion College in Zwolle to produce the animation, Richard van Kruysdijk for its translation into English and Tineke Bouwkamp-Timmer for editing.

## Author Contributions

**Conceptualization:** Irène Alders, Carolien Smits, Paul Brand, Sandra van Dulmen.

**Data curation:** Irène Alders.

**Formal analysis:** Irène Alders, Carolien Smits.

**Investigation:** Irène Alders.

**Methodology:** Irène Alders, Carolien Smits, Paul Brand, Sandra van Dulmen.

**Supervision:** Paul Brand, Sandra van Dulmen.

**Validation:** Carolien Smits.

**Writing – original draft:** Irène Alders.

**Writing – review & editing:** Carolien Smits, Paul Brand, Sandra van Dulmen.

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
