## [Decision Letter · Decision Letter 0]

17 Jan 2022

PONE-D-21-23544Patient coaching: what do patients want? A mixed methods study in waiting rooms of outpatient clinicsPLOS ONE

Dear Dr. Alders,

Thank you for submitting your manuscript to PLOS ONE. After careful consideration, we feel that it has merit but does not fully meet PLOS ONE’s publication criteria as it currently stands. Therefore, we invite you to submit a revised version of the manuscript that addresses the points raised during the review process.

Please carefully review and respond to the reviewers' comments as I believe they will improve the quality and clarity of the manuscript. Overall, I agree with the reviewers' comments, including revisions to the introduction which I think could be made more concise and clear. Further, please consider adding additional detail on strategies that were used to enhance the trustworthiness of the study (item 15 in the SRQR). If this is present and I missed it simply clarify where it is located. Should you disagree with any of the reviewers' comments please provide a rebuttal. Once these changes are responded to, we will reassess as to whether additional changes are needed prior to publication.  

We look forward to receiving your revised manuscript.

Kind regards,

Michael R. Gionfriddo, Pharm.D, Ph.D

Academic Editor

PLOS ONE

Journal Requirements:

2. Please provide additional details regarding participant consent. In the ethics statement in the Methods and online submission information, please ensure that you have specified whether consent was written or verbal/oral. If consent was verbal/oral, please specify: 1) whether the ethics committee approved the verbal/oral consent procedure, 2) why written consent could not be obtained, and 3) how verbal/oral consent was recorded. If your study included minors, please state whether you obtained consent from parents or guardians in these cases. If the need for consent was waived by the ethics committee, please include this information.

3. We note that video presented in the manuscript text line 182 contain copyrighted images/ videos. All PLOS content is published under the Creative Commons Attribution License (CC BY 4.0), which means that the manuscript, images, and Supporting Information files will be freely available online, and any third party is permitted to access, download, copy, distribute, and use these materials in any way, even commercially, with proper attribution. For more information, see our copyright guidelines: http://journals.plos.org/plosone/s/licenses-and-copyright.

a. You may seek permission from the original copyright holder of video presented in the manuscript text line 182 to publish the content specifically under the CC BY 4.0 license. 

Reviewers' comments:

Reviewer's Responses to Questions

**Comments to the Author**

1. Is the manuscript technically sound, and do the data support the conclusions?

Reviewer #1: Yes

Reviewer #2: Yes

2. Has the statistical analysis been performed appropriately and rigorously? 

Reviewer #1: Yes

Reviewer #2: Yes

3. Have the authors made all data underlying the findings in their manuscript fully available?

Reviewer #1: Yes

Reviewer #2: Yes

4. Is the manuscript presented in an intelligible fashion and written in standard English?

Reviewer #1: Yes

Reviewer #2: No

5. Review Comments to the Author

Reviewer #1: This manuscript describes the results of a mixed methods study addressing patient's interest in receiving assistance from a patient coach during a specialty visit in the Netherlands, the kind of assistance desired, and the characteristics of a coach that are important to patients. In particular, this study explores the characteristics of patients interested in receiving a patient coach. While there is growing evidence for the efficacy of coaching interventions to improve the patient experience, there is significantly less information about who desires patient coaching or what they are seeking from the experience.

The authors' findings that no demographic characteristics were significantly associated with interest in coaching, but rather that low perceived efficacy was the only variable showing a significant different between patients with and without interest in support is a significant contribution to the literature. From an operational perspective, most coaching programs seek to identify patients via algorithms that characterize their "risk" -- which rely heavily on age, chronic conditions, and indications of polypharmacy. This study challenges the assumptions underlying that algorithm, suggesting that the patient's perceived efficacy may actually be a better way to identify people who believe they could benefit from coaching support. It is also interesting that in the context of this setting, patients interested in coaching support desired the coach presence during the medical visit as an advocate. In the U.S., most coaching models do not include the presence of a health coach in the medical visit, so this is an interesting observation.

I have minor suggestions to the authors:

p.2, lines 42-43: "Subsequently, interested patients 43 were asked to participate in a semi-structured interview..." It is not clear in the abstract whether you mean patients interested in a coach or an interview.

Results: I would suggest a general revision of the methods section to first present response rate and characteristics of the entire sample, then to proceed to the main finding (regarding efficacy as the primary variable associated with desire to receive a patient coach).

The authors chose to dichotomize age. Could you confirm in text that results of the analysis are not different if age is used as a continuous variable?

Reviewer #2: Thank you for the opportunity to review this study. The authors conducted a strong mixed methods study. The methods, results, and discussion sections are particularly strong. Overall, the introduction section deserves the greatest attention to better set up the remainder of the paper. There are also several grammatical errors scattered throughout – particularly run on sentences – that I have tried to note where I observed them. Finally, in the discussion section (and possibly in the revised introduction as well) the authors should clearly distinguish patient communication coaches from other types of coaches such as Health and Wellness Coaches and Capacity Coaches.

Section by section notes are detailed below:

Abstract

In the methods subsection, “profound” seems to be a bit strong of a word.

Introduction

Vague in some areas. For example, the sentence “However, not all patients are able to communicate effectively in consultations with medical specialists (2-6)” has multiple citations but it is unclear to me what some of the, likely nuanced, reasons that patients may have difficulty in these situations. I think it is also important to note what aspects of communication are uniquely suited to a third person (the patient coach) vs improving communication skills on the part of the clinician.

What is meant by “personal” support? This word is italicized so presumably important, but I am unclear exactly what is meant by it.

Line 71 Shared decision making should be discussed in a separate sentence.

79 intervention should be interventions

117 This sentence points out the two comparison groups, but the following sentences don’t reference any of the results of the comparison. If the comparison is meaningful it should be discussed. Otherwise, I think reference to the comparison can be disregarded.

129 Revise sentence to: To shed more light on individual patients’ needs, we investigated the characteristics of patients that would like support from a patient coach when consulting a medical specialist and their reasons for desired support.

The research questions at the end of the introduction as a list feels informal. These should be incorporated into the introduction in paragraph form.

Methods

Mixed methods design should be referenced (explanatory sequential). See Creswell, designing and conducting mixed methods research for examples.

144-146 this sentence should be broken into two

148 should be Bachelor’s of Nursing

147 -150 run on sentence, edit/separate

Table of characteristics – for mutually exclusive categories (e.g. Male/Female as other gender identities do not appear in the table) only one line is required.

Low/high confidence in self-efficacy could be simplified to low/high self-efficacy.

Discussion:

I am still confused by what defines personal and non-personal which is used in the discussion section (line 469) as well as intro.

Line 479: Needs citation: “In approximately four out of five patient-specialist encounters patients did not get the opportunity to discuss their agenda.”

Line 520: The type of coaching described here may be more in alignment with Health and Wellness Coaching (see Wolever 2013) or Capacity Coaching (see Boehmer 2019). These coaches interface with patients outside of the medical encounter and support self-management and wellness behaviors. It may be important to note these as you distinguish them from a communication coach. It may even be useful to adopt the name of Patient Communication Coach to help distinguish future interventions from other coaching interventions, particularly as Health and Wellness Coaching is now a board-certified specialty in the United States.

6. PLOS authors have the option to publish the peer review history of their article (what does this mean?). If published, this will include your full peer review and any attached files.

Reviewer #1: No

Reviewer #2: **Yes: **Kasey R. Boehmer

---

## [Author Response · Author response to Decision Letter 0]

8 Mar 2022

In the uploaded rebuttal letter our answers to the remarks and questions of the reviewers and editor are underlined to distinguish them. 

Patient coaching: what do patients want? A mixed methods study in waiting rooms of outpatient clinics

Irène Alders 1, CarolienSmits 2, Paul Brand 3,4, Sandra van Dulmen 1,5

Editor:

Dear dr. Gionfriddo, 

Thank you very much for considering publication of our paper. We trust we adjusted the text according to your requirements and the suggestions for improvement. 

Looking forward to your positive decision,

Kind regards, on behalf of all co-authors,

Irène Alders

P.S. The lines refer to the lines in the marked up version of the revised manuscript.

Overall, I agree with the reviewers' comments, including revisions to the introduction which I think could be made more concise and clear. 

1) Trustworthiness. Further, please consider adding additional detail on strategies that were used to enhance the trustworthiness of the study (item 15 in the SRQR). If this is present and I missed it simply clarify where it is located. 

We added information on trustworthiness in the Methods section:

Lines 213 – 216: “The interviews were audio-recorded, transcribed verbatim by the nursing students and subsequently all checked for accurate transcription by the first author. The transcripts were analyzed through content analysis [22].”

2) In the ethics statement in the Methods and online submission information, please ensure that you have specified whether consent was written or verbal/oral. If consent was verbal/oral, please specify: 1) whether the ethics committee approved the verbal/oral consent procedure, 2) why written consent could not be obtained, and 3) how verbal/oral consent was recorded. If your study included minors, please state whether you obtained consent from parents or guardians in these cases. If the need for consent was waived by the ethics committee, please include this information. 

To answer your questions: 

1) All participants in the survey provided written informed consent. For the interviews one participant provided only oral consent, after having received all information in writing. No specifics on how oral consent should be recorded was provided in the METC approval, but we audio recorded the consent at the beginning of the interview. 

2) We described why one participant was interviewed by telephone.

3) We added how the consents were recorded: written or by audio record. 

The text in this paragraph was adjusted as follows:

Lines 164 – 172: “Their participation was voluntary, and they could withdraw at any time. All participating patients in the survey provided informed consent. For the interviews, seven patients provided written consent. The interview with one patient was cancelled in agreement with the patient’s partner, because of the patient’s cognitive condition. A day later this interview was continued by his partner by telephone in which additional oral consent was obtained and audio recorded. The eighth participant preferred to participate in the interview by telephone. This informed consent was obtained orally and audio recorded.”

3) We require other proof of granted permissions as an "Other" file with your submission. 

We added the content permission form to the submission. In 2018 I (I, Irène Alders, first author) commissioned Multimedia students from the Deltion college in Zwolle to create an animation on patient coaching as part of their internship. I am the owner of this animation. At the time, correspondence went through my email account at Windesheim University of Applied Sciences, where I worked as a lecturer. Currently, I no longer work there and have no access to that account anymore. On Youtube I have added information about the usage rights of the animation (CC BY 4.0 license).

This now reads as: 

Lines 202 – 203: “….animation (https://youtu.be/iF4kkHG2l2M) (Reprinted from Youtube under a CC BY license, with permission from Irène Alders, original copyright 2018).”

Reviewer #1: 

Thank you for your kind remarks on the results of the study and your suggestions to improve clarity. We underlined the adjustments made to the text.

I have minor suggestions to the authors:

p.2, lines 42-43: "Subsequently, interested patients 43 were asked to participate in a semi-structured interview..." It is not clear in the abstract whether you mean patients interested in a coach or an interview.

Adjusted: Lines 34 – 35: “Subsequently, patients interested in a patient coach were asked.”

Results: I would suggest a general revision of the methods section to first present response rate and characteristics of the entire sample, then to proceed to the main finding (regarding efficacy as the primary variable associated with desire to receive a patient coach).

We took this reviewer’s comment to mean that the results section needed a general revision instead of the methods section. With all due respect to the reviewer, we believe that we described the results as requested, see lines 240 – 262. Perhaps we did not understand this point correctly. If so, would you be so kind as to clarify your point?

To focus on the main findings in the discussion section we added the following lines:

Lines: 438 – 440): “Perceived efficacy in patient-physician interactions was the only variable that showed a significant difference between patients with and without an interest in support from a patient coach.”

The authors chose to dichotomize age. Could you confirm in text that results of the analysis are not different if age is used as a continuous variable?

We performed an additional analysis and changed the text as follows: 

Lines 254 – 257: “We did not identify any other factors that were associated with an interest in patient coaching. The results did not differ if age was used as a continuous variable (p=0.156, Pearson Chi Square, 2 sided, range 19-87 years (Table 1).” 

Reviewer #2: 

Dear dr. Boehmer, 

Thank you for your time and remarks on our study. We appreciate your suggestions very much. Below we describe the adjustments we made accordingly. 

Overall, the introduction section deserves the greatest attention to better set up the remainder of the paper. 

According to the reviewer’s remark we adjusted the introduction section. 

There are also several grammatical errors scattered throughout – particularly run on sentences – that I have tried to note where I observed them. 

We carefully reviewed the manuscript text for any remaining grammatical errors.

Finally, in the discussion section (and possibly in the revised introduction as well) the authors should clearly distinguish patient communication coaches from other types of coaches such as Health and Wellness Coaches and Capacity Coaches.

In the introduction section we focused more on the aim of patient coaching to coach patient to communicate effectively in consultations with medical specialists. This part of the introduction now reads as:

Lines 73 – 75: “To support patients in communicating effectively during these consultations, several guiding and coaching interventions for patients have been developed and investigated [8, 9].”

In the discussion section we started with the sentence: “In this study we investigated the views of patients in the waiting room of outpatient clinics on patient coaching to support effective communication in consultations with a medical specialist.” With the forementioned adjustment in the introduction we trust this will point out the focus of patient coaching on communication and distinguish it from Health and Wellness coaches and Capacity coaches.

Below, you can find our detailed responses (underlined) to each of the points made by the reviewer:

Section by section notes are detailed below:

Abstract

In the methods subsection, “profound” seems to be a bit strong of a word.

Adjusted: 

Line 30: “We applied a mixed method design to obtain a realistic understanding of patients’ perspectives on a patient coach.”

Introduction

Vague in some areas. 

For example, the sentence “However, not all patients are able to communicate effectively in consultations with medical specialists (2-6)” has multiple citations but it is unclear to me what some of the, likely nuanced, reasons that patients may have difficulty in these situations. 

We added a sentence with example barriers from these references and trust this adds to the clarity. 

This now reads as: 

Lines 65 – 70: ”This is caused by the patient’s emotional state, like feeling tense or overwhelmed, the felt time pressure, uncertainty about their own understanding, not wanting to be bothersome, remembering questions only after the consultation and also the attitude of the professional [2-4]. Furthermore, patients are hindered by the power imbalance [5], or their inability to change the agenda in the consultation [6].” 

I think it is also important to note what aspects of communication are uniquely suited to a third person (the patient coach) vs improving communication skills on the part of the clinician. 

To clarify the added value of a patient coach as a third party we added the following sentences:

Lines 70 -73: “Although medical specialists are increasingly trained in communication skills, transfer to real consultations is still limited [7]. Furthermore, the consultation time remains limited, and training medical specialists does not solve the experienced power imbalance.”

What is meant by “personal” support? This word is italicized so presumably important, but I am unclear exactly what is meant by it. 

We added information on the value of personal support. As non-personal support, one could, for instance, think of online self-management support interventions.

We made adjustments in the text, trusting this will clarify both points. 

The text now reads as: 

Lines 75 – 84: “It appears that personal, face-to-face support may be best suited. The human connection is invaluable in the context of person-centered care and helps to make patients feel respected and equal [10]. 

When a patient coach spends time with a patient in preparation of the consultation(s), he gets to know the patient in his own context. During the accompanied consultations, the patient’s specific communication barriers are enlightened and can be addressed by the patient coach. Personal support can easily and instantly be adjusted to better meet an individual patient’s circumstances and needs [2].” 

Line 71 Shared decision making should be discussed in a separate sentence.

Added: 

Lines 63 – 64: “In shared decision making, the contribution of a patient is essential.”

79 intervention should be interventions

Adjusted (Line 87)

117 This sentence points out the two comparison groups, but the following sentences don’t reference any of the results of the comparison. If the comparison is meaningful it should be discussed. Otherwise, I think reference to the comparison can be disregarded.

Adjusted: 

Lines 125 – 126: “So far, research on patient coaches has shown that they have various backgrounds, ranging from lay educators to trained professionals, but a relationship between the coach’s profile and goals or outcomes of the coaching has not been investigated [8].”

129 Revise sentence to: To shed more light on individual patients’ needs, we investigated the characteristics of patients that would like support from a patient coach when consulting a medical specialist and their reasons for the desired support.

Adjusted according to the suggestion (Lines 139 – 142)

The research questions at the end of the introduction as a list feels informal. These should be incorporated into the introduction in paragraph form.

Paragraph adjusted according to the suggestion which now reads as: 

Lines: 142 – 145: “Our research questions were: which patients are interested in support from a patient coach, how should a patient coach support a patient, and what characterizes the ideal patient coach?”

Methods

Mixed methods design should be referenced (explanatory sequential). 

See Creswell, designing and conducting mixed methods research for examples.

Adjustment: 

Lines 151 – 155: “This mixed methods study comprised a survey amongst patients in an outpatient clinic waiting room and subsequent in-depth interviews with a sample of the survey respondents who had indicated an interest in a patient coach in the survey[19]. A mixed methods design was chosen to obtain a more profound understanding of patients’ perspectives on support from a patient coach.”

A reference of Creswell (2019) on mixed methods studies was added. 

144-146 this sentence should be broken into two

Adjusted according to the suggestion. The sentences now read as: 

Lines 158 – 161: “We invited 203 patients in the waiting room of outpatient clinics for chronic diseases (cardiology, pulmonology, rheumatology, oncology, internal medicine, and geriatrics) to participate in our study. In these clinics, we were likely to encounter vulnerable patients.”

148 should be Bachelor’s of Nursing

Adjusted in line 162.

147 -150 run on sentence, edit/separate

Adjusted, the sentences were split and now read as: 

Lines 161 – 163: “Prior to a consultation with a medical specialist, two Bachelor’s of Nursing students informed the patients about the objectives and procedures of the study. They explained the concept of patient coaching and asked the patients to participate in the survey.”

Table of characteristics – 

o for mutually exclusive categories (e.g. Male/Female as other gender identities do not appear in the table) only one line is required.

o Low/high confidence in self-efficacy could be simplified to low/high self-efficacy.

• Male/Female: We agree with the reviewer that one line would suffice, but since we also wanted to present whether there was any significant difference in interest in a patient coach between males and females, we prefer to keep the description in two lines. 

• We simplified the low/high self efficacy lines in the table according to the suggestion. 

Discussion:

I am still confused by what defines personal and non-personal which is used in the discussion section (line 469) as well as intro.

We trust our additional explanation clarifies the definition of patient coaching.

In the introduction we adjusted the text as follows: 

Lines 72 – 80: ”It appears that personal, face-to-face support may be best suited. The human connection is invaluable in the context of person-centered care and helps to make patients feel respected and equal [9]. When a patient coach spends time with a patient in preparation of the consultation(s), he gets to know the patient in his own context. During the accompanied consultations, the patient’s specific communication barriers are enlightened and can be addressed by the patient coach. Personal support can easily and instantly be adjusted to better meet an individual patient’s circumstances and needs [2].”

In the discussion it now reads as:

Lines 485 – 487: “This support should be given in person, because a process of human connectedness appears to be more valued by patients, than the provision of written or online information only [32].”

Line 479: Needs citation: “In approximately four out of five patient-specialist encounters patients did not get the opportunity to discuss their agenda.”

(lines 507 – 50) This was the same reference as used in the next sentence, but we followed your suggestion and added the reference after this sentence as well. (Reference 36) 

Line 520: The type of coaching described here may be more in alignment with Health and Wellness Coaching (see Wolever 2013) or Capacity Coaching (see Boehmer 2019). 

These coaches interface with patients outside of the medical encounter and support self-management and wellness behaviors. It may be important to note these as you distinguish them from a communication coach. It may even be useful to adopt the name of Patient Communication Coach to help distinguish future interventions from other coaching interventions, particularly as Health and Wellness Coaching is now a board-certified specialty in the United States.

We agree that it is important to distinguish between types of coaching. We are aware of various types of coaching and that this might be confusing to patients. The naming of this supportive intervention is a delicate matter and we are considering alternatives. Adding “communication” might trigger resistance in patients who need a patient coach, since they think of themselves as effective communicators on forehand. To find the most suited name, it might even be best to consult potential users. Because in our previous studies, we used the term “patient coach” for this intervention, we prefer to continue to use this term in the present manuscript.

---

## [Decision Letter · Decision Letter 1]

20 Apr 2022

PONE-D-21-23544R1Patient coaching: what do patients want? A mixed methods study in waiting rooms of outpatient clinicsPLOS ONE

Dear Dr. Alders,

Thank you for submitting your manuscript to PLOS ONE. After careful consideration, we feel that it has merit but does not fully meet PLOS ONE’s publication criteria as it currently stands. Therefore, we invite you to submit a revised version of the manuscript that addresses the points raised during the review process.

The reviewer requires one additional clarification prior to it being acceptable to publish. 

We look forward to receiving your revised manuscript.

Kind regards,

Michael R. Gionfriddo, Pharm.D, Ph.D

Academic Editor

PLOS ONE

Journal Requirements:

Reviewers' comments:

Reviewer's Responses to Questions

**Comments to the Author**

1. If the authors have adequately addressed your comments raised in a previous round of review and you feel that this manuscript is now acceptable for publication, you may indicate that here to bypass the “Comments to the Author” section, enter your conflict of interest statement in the “Confidential to Editor” section, and submit your "Accept" recommendation.

Reviewer #2: (No Response)

2. Is the manuscript technically sound, and do the data support the conclusions?

Reviewer #2: Yes

3. Has the statistical analysis been performed appropriately and rigorously? 

Reviewer #2: Yes

4. Have the authors made all data underlying the findings in their manuscript fully available?

Reviewer #2: Yes

5. Is the manuscript presented in an intelligible fashion and written in standard English?

Reviewer #2: Yes

6. Review Comments to the Author

Reviewer #2: Thank you for your revised version of the manuscript. I have found all of the authors' modifications to be acceptable, with one minor exception that should still be addressed.

While I do agree it is fine to keep the language regarding patient coach instead of patient communication coach, there needs to be a more clear note that this is distinctly different from Health and Wellness Coaching. The reason I am being very specific about this point is because that in the US context (recognizing this is different than the primary study context), services from a Board Certified Health and Wellness Coach are a billable and reimbursable service by some insurances. The only way to receive reimbursement is for the service to be performed by a Board Certified Health and Wellness Coach, and so we want to ensure that readers do not confuse this intervention with the reimbursable service. For more information regarding certified HWC, please see https://nbhwc.org/.

7. PLOS authors have the option to publish the peer review history of their article (what does this mean?). If published, this will include your full peer review and any attached files.

Reviewer #2: **Yes: **Kasey R. Boehmer

---

## [Author Response · Author response to Decision Letter 1]

28 Apr 2022

Dear mrs. Boehmer,

Thank you for your remark. We added information which distinguishes a patient coach from a Board Certified Health and Wellness coach in the abstract and in the text. We trust these adjustments are acceptable to you.

The abstract now reads as:

Lines 26 -27: “These patients could benefit from support from a coach who accompanies them to and during medical specialist consultations to improve communication in the consultation room.”

The text in the introduction now reads as:

Lines 86 – 90: “We defined the concept of patient coaching as personal support for patients, aiming at improving communication in consultation with a medical specialist. The patient is supported in the preparation of the consultation, accompanied during the consultation and in the evaluation of the consultation with a medical specialist afterwards.”

Kind regards, 

On behalf of my co-authors,

Irène Alders

---

## [Editor Report · Decision Letter 2]

26 May 2022

Patient coaching: what do patients want? A mixed methods study in waiting rooms of outpatient clinics

PONE-D-21-23544R2

Dear Dr. Alders,

We’re pleased to inform you that your manuscript has been judged scientifically suitable for publication and will be formally accepted for publication once it meets all outstanding technical requirements.

Kind regards,

Michael R. Gionfriddo, Pharm.D, Ph.D

Academic Editor

PLOS ONE

Additional Editor Comments (optional):

The article is now acceptable for publication. Congratulations
---

## [Editor Report · Acceptance letter]

31 May 2022

PONE-D-21-23544R2 

Patient coaching: what do patients want? A mixed methods study in waiting rooms of outpatient clinics 

Dear Dr. Alders:

I'm pleased to inform you that your manuscript has been deemed suitable for publication in PLOS ONE. Congratulations! Your manuscript is now with our production department. 

Kind regards, 

on behalf of

Dr. Michael R. Gionfriddo 

Academic Editor

PLOS ONE